# The Global Burden of Disease of Zoonotic Parasitic Diseases: Top 5 Contenders for Priority Consideration

**DOI:** 10.3390/tropicalmed4010044

**Published:** 2019-03-02

**Authors:** Konrad Pisarski

**Affiliations:** 1Division of Tropical Health & Medicine, College of Public Health, Medical and Veterinary Sciences, James Cook University, Townsville, QLD 4814, Australia; konrad.pisarski@my.jcu.edu.au; Tel.: +04-0080-0576; 2Faculty of Medicine, University of Queensland, Herston, St Lucia, QLD 4072, Australia

**Keywords:** cryptosporidiosis, intestinal nematode infections, leishmaniasis, schistosomiasis, lymphatic filariasis, burden of disease, parasites, zoonoses, neglected tropical diseases

## Abstract

With the rise of global migration, international trade, and global environmental challenges such as climate change, it is not surprising that the interactions between humans and other animals are shifting. Salient infectious diseases, such as malaria and HIV (which have high burdens of disease), attract sophisticated public health frameworks and funding from global/regional organisations, such as the WHO. This unfortunately detracts attention from the many emerging zoonoses that fall under the radar as neglected tropical diseases (NTDs). This review considers the available literature and the attribution of burden of disease to the most insidious NTDs and recommends which five are deserving of policy prioritisation. In line with WHO analyses of NTDs, intestinal nematode infections, leishmaniasis, schistosomiasis, and lymphatic filariasis should be prioritised, as well as the burden of disease of cryptosporidiosis, which is largely underestimated. Both monitoring and treatment/prevention control methods for cryptosporidiosis are suggested and explored.


**The Global Burden of Disease of Zoonotic Parasites: Top Five Contenders for Priority Consideration.**


When considering global burden of disease (BOD) of infectious origin, it is usually certain ‘poster-child’ diseases that come to mind, such as tuberculosis, malaria, avian influenza, and HIV. This is reflected in global prevention strategies and allocation of resources [1]; however, it also means that there is a large body of infectious diseases called neglected tropical diseases (NTDs) that may fall below the global public health radar and either continue untreated or pose possible pandemic risk [2]. Specifically, zoonotic parasitic infections are of interest, considering the shifting interactions between humans and other animals as well as global trade and agriculture—with the rise of One Health as a public health discipline within the past decade [3,4]. This literature review will consider emerging parasitic zoonoses that fall into the category of NTDs and contribute the most to the global BOD. Upon outlining the relevant literature and presenting the top five zoonotic parasitic diseases in terms of global BOD, an integrated control program will be suggested for cryptosporidiosis, which is largely neglected in salient public health literature and public health policy.

On the topic of emerging parasitic zoonoses, a piquing observation was made by Thompson and Deplazes (2011): ‘In the context of emerging diseases, parasite zoonoses figure prominently, undoubtedly because of the shifting interactions between humans and other animals.’ [5]. Regarding the sheer number of emerging infectious diseases, the use of ‘prominently’ is definitely accounted for, with over 60% of around 400 identified infectious diseases since 1940 being zoonotic [4]. However, it is important to view this in light of global BOD (which is often underestimated for these diseases) and the relative methods of transmission of these zoonoses. For example, certain zoonotic parasites may affect humans as incidental hosts (e.g., toxoplasmosis), where infection is dependent upon human behaviour/interaction with the definite hosts, while other zoonotic parasites readily infect humans and are more easily spread through environmental modes such as rain runoff, flooding, vectors, and aerosol spread [6].

For the purposes of this review, the global BOD framework proposed by Hotez et al. (2014) [7] will be used to explore the top five parasitic zoonoses (Table 1): (i) cryptosporidiosis; (ii) intestinal nematode infections (INI); (iii) leishmaniasis; (iv) schistosomiasis; and (v) lymphatic filariasis (LF). Despite listing BOD calculations in terms of disability adjusted life years (DALYs), which can further be broken down into components of years lived with disability (YLD) and years lost to premature mortality (YLL) [8], the analysis done by Hotez et al. (2014) through the World Health Organization (WHO) is not without its criticisms [7]. Briefly, criticisms of the analysis include existing gaps in data (especially in poorer parts of the world) and estimates of population health being prone to differing cultural frameworks—somewhat obscuring the extent of the BOD due to a given disease [9]. A full critique of that analysis is beyond the scope of this review; however, what remains important to remember is the economic limitations of instituting public health prevention strategies on a global scale and how resource management is necessary when dealing with a largely neglected subset of global infectious diseases [9].

## 1. Cryptosporidiosis

Possibly the largest critique of the WHO’s analysis of NTDs is the choice to not include cryptosporidiosis in the NTD category despite it contributing significantly to the global BOD, with an estimate of 8.37 million DALYs according to 2010 data, which may be an underestimation in itself [7]. Recent evidence suggests that in the developed world, cryptosporidium diagnoses may reflect somewhere as low as 1% of true prevalence, with this being even lower in the developing world [10]. Considering the global distribution of protozoa of the *Cryptosporidium* genus and that immune-compromised patients and children are at risk of developing cryptosporidiosis, this suggests a greater BOD when paired with its diagnostic evasiveness [11]. In recent studies, estimates show that up to a quarter of children with diarrhoea are co-morbidly infected with cryptosporidium, which is associated with longer durations of diarrhoea, malnutrition, higher mortality rates, and immune-compromise [12].

According to the Centers for Disease Control and Prevention (CDC), transmission of *C. parvum* and *C. hominis* (the two primary species affecting humans) occurs mainly through drinking or eating contaminated water/food where thick-walled oocysts are ingested, allowing further parasite maturation in the intestines and reproduction into auto-infective thin-walled oocysts and thick-walled oocysts that pass through faeces into the environment [13]. Interestingly, this life cycle is not unique to humans with other *Cryptosporidium* spp. infecting essentially all mammals, including those in close domestic proximity to humans [13]. However, according to a recent study, no significant difference in BOD was found between households that used bottled water and those that used tap water, suggesting that the primary method of spread lies elsewhere [14]. When paired with higher levels of oocysts (which are quite resistant to chlorination attempts) in surface water at the end of rainy seasons, seasonal patterns in cryptosporidiosis suggest that rainwater-runoff in areas with poor infrastructure are perfect breeding grounds for *Cryptosporidium* spp. due to the hot and humid environment, so there is an increased risk of infection [15]. There is much we do not know about the extent of cryptosporidiosis globally; however, if the BOD is to be lowered and improved, accessible diagnostic methods must be made available globally (with animals also tested) [11]. Treatments targeting the reasons for infection, such as immunocompromised/AIDS or INIs, should also be prioritised and supplemented with antiparasitic therapy [11].

## 2. Intestinal Nematode Infections (INI)

DALY analyses of the global BOD of NTDs have systematically demonstrated the dominance and wide epidemiological spread of INIs with an estimated 5.19 million DALYs [7]. Of these, the majority are accounted for by hookworm diseases (3.23 million), followed by ascariasis (1.32 million), then by trichuriasis (0.64 million) [7]. Interestingly in the WHO analysis, strongyloidiasis was not included in the INI figure, which may result in a severe underestimation of the total BOD arising from nematodes. Regarding global prevalence, INIs are most prevalent in Asia with roughly 67% of cases; however, incidence rates controlled for country population are quite unvaried between major global regions [15]. The burden of INIs seems to vary greatly within major global regions, which makes sense considering the methods by which these diseases are spread—water runoff, rain, animals, and human migration amidst inadequate public sanitation and infrastructure [16].

The lifecycle of soil-transmitted helminths is important to consider, as the eggs of the helminth are passed from the definitive host into the environment at the infective stage [11,16,17]. The larvae of *Strongyloides stercoralis*, unlike other soil-transmitted helminths, are passed in faeces at the infective stage [11,16,17]. The pathological sequelae of *Ascaris lumbricoides* and *Trichuris trichiura* primarily involve symptomatic infection (and associated immunocompromise), wasting, and abdominopelvic problems, whereas the added complication of anaemia (from bleeding) is present in hookworm infections [17]. Maturation of eggs/larvae in the environment differ between helminthic species with some species requiring eggs to hatch into larvae (non-infective eggs) and others requiring the ingestion of eggs for maturation into adult worms; what remains constant is the need for soil/environmental infestation of eggs/larvae in order to complete the life cycle [17]. Eggs can survive from months to years in the environment, and with the addition of mammalian hosts the spread of a wider variety of zoonoses is made possible, with humans being incidental hosts of nematodes such as in the life cycle of *Trichinella* spp. [17].

## 3. Leishmaniasis

Leishmaniasis accounts for a significant proportion of global BOD from NTDs with an estimate of 3.32 million DALYs according to Hotez et al. (2014) [7]. Most tropical countries are at risk of endemic leishmaniasis with spread being likely, considering that the primary vector is the phlebotomine sandfly, which has over 30 species and does not respect international country borders [18]. A large problem in *Leishmania* spp. endemic countries is co-infection with other NTDs, such as HIV, which act together to produce immunocompromise and increased levels of morbidity associated with leishmaniasis [18]. Much like *Anopheles* spp. malaria vectors, phlebotomine sandfly distribution is limited to areas above a certain temperature (in this case 15.6 °C), and with global warming ultimately pushing the latitudinal tropical borders further apart, the risk of spread of leishmaniasis via sandflies to naïve countries is a real possibility [19].

The most common form of leishmaniasis is cutaneous and presents with skin sores, which can be painful and ultimately result in debilitating social isolation and decreased quality of life (sometimes leading to mucosal leishmaniasis) similar to lymphatic filariasis [20]. Visceral leishmaniasis, also significantly present in the tropics, has a much higher morbidity rate than cutaneous resulting in multiple organ failure as well as anaemia, leukopenia, and thrombocytopenia [20]. Regarding the spread of leishmaniasis, depending on the region of the world and strain of *Leishmania* spp. the parasitic life cycle can be propagated without the presence of humans, with rodents and dogs being able to act as reservoirs for the disease [20,21]. In regions where anthroponotic spread is prevalent, treating individuals will help break the cycle of spread; however, with changing climates, increasing human traffic, and increases in global population density, the prevalence of less common forms of leishmaniasis without the need for anthropogenic spread could increase [20,21].

## 4. Schistosomiasis

Although schistosomiasis has a similar global BOD, as estimated by Hotez et al. (2014) at 3.31 million DALYs, it has a significantly higher proportion of YLL % and higher global prevalence [7]. *Schistosoma japonicum, S. haematobium*, and *S. mansoni* are the primary pathogens of humans contributing to the majority of the global BOD. The geographic distribution lies primarily in Africa/Caribbean/Middle East for *S. mansoni* and *S. haematobium*, and in Asia for *S. japonicum*. In addition, the salient human schistosomes *S. intercalatum* and *S. mekongi* are primarily found in Central-West Africa and Southeast Asia, respectively [22,23]. The vectors for all *Schistosoma* spp. are types of snails, which are infected by *Schistosoma* spp. miracidia that hatch in water following human defecation or urination (for *S. haematobium*). These further develop within the snail into cercariae (the infective forms), which are then released back into the water and directly infect the definitive hosts when they comes into contact with them [22,23]. Of significance is the ability of *S. mekongi* and *S. japonicum* to live in animal reservoirs including many agricultural/domestic animals such as dogs, cats, pigs, goats, and horses [22,23].

With the increasing reliance on local agricultural produce and a lack of adequate infrastructure and sanitation in parts of the globe most affected by schistosomiasis, the threat of spread of the disease should be considered with surveillance/monitoring of infection in populations, snail xenodiagnoses, and adequate treatment regimens [22,24]. There are two main stages of schistosomiasis—the chronic and acute stages—with the acute stage usually presenting in pathogen-naïve persons as a cluster of symptoms known as Katayama syndrome [22,23]. Over time, the worms take refuge in the mesenteric venules/hepatic portal system, causing crippling pathological sequelae ranging from organomegaly/cirrhosis to gastrointestinal bleeding (haematuria for *S. haematobium*) and anaemia [22,23]. A hallmark feature of schistosomiasis often overlooked by BOD estimates is the long-lasting effects of the illness on persons even long after de-worming [22,25]. Growth stunting, cognitive impairment, and permanent organ damage/increased risks of numerous cancers all add up to nearly double the global BOD for an already underestimated disease, according to Colley et al. (2014) [22,23,25].

## 5. Lymphatic Filariasis (LF)

At the bottom of the top five list by global BOD, as calculated by Hotez et al. (2014), is LF—accountable for approximately 2.78 million DALYs, mostly through YLDs [7]. Although the three pathogenic helminths responsible for LF, *Brugia malayi, Brugia timori*, and *Wuchereria bancrofti*, are technically nematodes, the fact that they are transmitted by mosquitos at the larval stage warrants their separate treatment to the majority of soil-transmitted nematodes [26]. The disease is found within the tropics, specifically in East/Southeast Asia, Oceania, Africa, and South America, with many different mosquito species able to function as vectors including *Culex* spp., *Anopheles* spp., *Aedes* spp., *Mansonia* spp., and *Coquillettidia juxtamansonia* [26,27]. This wide range of possible vectors brings with it a very salient risk of future LF outbreaks and spread beyond the currently endemic areas if proper public health systems and prophylactic measures are not taken to control spreading vectors (due to urbanisation and climate change) and prevent infection of at-risk persons [27].

The main pathophysiology of LF is linked to the L3 stage of the worm larvae, which further develop into adults and reside in the human lymphatics leading to lymphoedema, hydrocele, and an immunocompromised status resulting in increased opportunistic infections (which leave fibrotic tissue and thus elephantiasis), the majority of which account for the global BOD [26,27]. LF, unlike the other diseases mentioned in this review, quite rarely presents acutely, with the vast majority of sufferers living with chronic LF. However, the majority of the burden stems from the decrease in quality of life through psychiatric, social, and chronic suffering complications associated with the infection [7,26]. This is a double-edged sword, as although the rate of morbidity of LF is not high, this in turn means that governments and relevant stakeholders are less likely to prioritise LF treatment when other more apparent diseases, such as malaria and even leishmaniasis or schistosomiasis are present.

## 6. Control Methods for Cryptosporidiosis

Having considered five of the NTDs responsible for the greatest global BOD that should be prioritised by global stakeholders, the WHO, and nation-states, it is vital to consider the pragmatism of implementing any control or prevention strategies to curb the associated BOD. For example, LF elimination programs that originated as early as the year 2000 are already underway with mass drug administration and DEC-medicated salt as major programs for elimination [27]. The other zoonoses mentioned (leishmaniasis, schistosomiasis and INIs) currently all fall within the WHO’s NTD framework and coordinated multinational approaches to control these are either in place (de-worming strategies for INIs) or are planned for the near future [7,11]. What is key about the existing NTD framework, is that it inherently acts to prioritise certain diseases within the public health sphere for monitoring and surveillance, without which even the most effective treatment/control strategies would not be effective [11]. As such, cryptosporidiosis control and strategy development should be prioritised within the existing WHO NTD framework.

As outlined above, the detection of cryptosporidial infection is difficult for a number of reasons, many of which are caused by or compounded by low-resource settings in the most afflicted countries. The following domains of infection control will be outlined for cryptosporidiosis with recommendations made at the end (Table 2): (i) Monitoring/surveillance; (ii) treatment and prevention.

## 7. Monitoring/Surveillance

As previously mentioned, even with the best technology available the vast majority of cryptosporidium infections (which may be subclinical) go undetected, lying dormant until the host’s body is immunocompromised enough for an acute infection [11,28]. Detection of *Cryptosporidium* spp. can be done through four main methods including microscopy, antigen detection, nucleic acid amplification (ANA), and serological methods of stool samples; however, usually only microscopy is available in low-resource settings [29,30]. Considering the relative low sensitivity (70–80% with modified acid-fast stains) and human error component of microscopy, this becomes a problem when public health policy and control methods are based on estimates using this technology [11,30]. In Europe/USA, immunofluorescence microscopy, as well as ANA, are used quite readily; however, serological assays or PCR (which are more cost-laden) are a must when it comes to accurate epidemiological analysis of the global cryptosporidium problem [11,30]. Without adequate economic and socio-political investment in monitoring cryptosporidiosis, much progress will not be made, therefore it is recommended that national data are kept updated, with mandatory reporting of cryptosporidiosis by health practitioners.

The primary methods of spread of *Cryptosporidium* spp. is through the pathogens living in bodies of water that then come into contact with humans through direct ingestion by drinking or washing food with infected water [13]. In fact, there does not seem to be an increased risk of acquiring cryptosporidiosis from municipal drinking water—which goes through sterilization processes and pathogen testing—suggesting that the parasites of interest are found elsewhere [19]. The seasonal patterns of cryptosporidiosis also suggest that rainfall/flooding play a large part in the spread of the disease, with mammal reservoirs playing a large part in the spread [11,13,14]. What is worrying about these trends is the relative lack of consensus regarding the spread of *Cryptosporidium* spp., with xenodiagnoses being almost non-existent in low-mid income countries [11]. In terms of improving surveillance/monitoring of cryptosporidiosis globally, the first step would be to include this disease within the WHO’s Global Burden of Disease framework [7] to increase international awareness of the BOD associated with *Cryptosporidium* spp. Furthermore, xenodiagnosis has been a somewhat neglected tool for the prevention of protozoan spread and should be conducted whenever possible before animals are transported throughout agricultural trade [11]. This extends to the proper quarantine of food products for import/export and cooking food whenever possible, especially considering that *Cryptosporidium* spp. can survive cold temperatures, such as in refrigerators for up to several weeks [11,31].

## 8. Treatment and Prevention

Assuming adequate monitoring systems are in place for cryptosporidiosis, public health systems at a national level can react in one of two ways: (a) treatment (generally a short-term solution) and/or (b) prevention.

(a) Treatments in the form of antiparasitic medications are usually not effective for populations most at risk of contracting cryptosporidiosis *en masse*, such as those with HIV comorbidities or the elderly. As such, treatments for cryptosporidiosis tend to be complicated and vary from patient to patient, making successful public health interventions such as the LF elimination MDAs less likely to succeed [11,32,33]. For non-HIV infected patients, nitazoxanide has been shown to be an effective treatment for cryptosporidiosis and is relatively cheap at roughly $US10 per prescription dose [11,32,33]. However, it is the HIV-infected patients and subclinically infected children for which no real effective treatment has been developed [11]. Further research into novel drugs suitable for use in infants and HIV-immunocompromised patients is vital if any effective mass treatment initiatives are to take place to reduce the global BOD of cryptosporidiosis [11,34].

(b) Currently a vaccine for cryptosporidiosis does not exist; however, several factors have been identified suggesting that this would be a possibility due to the involvement of innate and adaptive immune processes including: (a) persons in endemic areas being less prone to re-infection; (b) prior infection requiring a higher infectious dose for re-infection; (c) increased severity of infection in immunocompromised patients [11,35]. At present, vaccine development is underway; however, it is likely to take a substantial amount of time considering the sheer number of antigens required for identification, as well as the relative immunological individuality of different *Cryptosporidium* spp. [11,35].

Ultimately, regardless of whether monitoring or treatment/prevention is being considered, more global funding and attention is required in order to realize any of these control strategies. The phenomenal cooperation of albendazole/ivermectin drug manufacturers with not-for-profits and state actors in the elimination of LF is a great model for diseases such as cryptosporidium, and as such, primary focus should fall on raising awareness of cryptosporidium as a global problem that could affect those in the developed world through food/agricultural trade [27]. The zoonotic aspects of cryptosporidiosis cannot be ignored, and further cooperation between public health professionals and those in agricultural/veterinary sectors should be encouraged.

## Figures and Tables

**Table 1 tropicalmed-04-00044-t001:** Summary of burden of disease (BOD) floor estimates of 5 most salient neglected tropical zoonotic parasitic diseases.

Rank	Disease	BOD (million DALYs)
1	Cryptosporidiosis	8.37
2	Intestinal nematode infections	5.16
3	Leishmaniasis	3.32
4	Schistosomiasis	3.31
5	Lymphatic filariasis	2.78

**Table 2 tropicalmed-04-00044-t002:** Summary of control methods for cryptosporidiosis [14].

Control Methods for Cryptosporidiosis		
Type of control method	Method used	Notes
Monitoring/Surveillance	Microscopy	Low sensitivity, but often the only option in low-resource settings. Sensitivity significantly higher with fluorescent antibody staining.
	Antigen detection	Expensive for poorer countries, however readily available commercial test kits with reasonable (70%+) sensitivity.
	Nucleic acid amplification (ANA)	Good for ruling out infection, however a low positive predictive value renders it not suitable as a first line for diagnosis.
	Serology	Limited to laboratories with no commercial kits available—useful for population surveillance of cryptosporidiosis.
	Xenodiagnosis	Underutilised method of surveillance which may be useful in areas suspected of harbouring disease vectors but are lacking in clinical/epidemiological surveillance of zoonotic disease.
Treatment/Prevention	Nitazoxanide	Useful for treatment of cryptosporidiosis in non-HIV patients, however it has limited usefulness for immunocompromised patients.
	*Cryptosporidium* vaccine	Evidence suggests development of a vaccine is possible, however it will likely take a significant amount of time to complete.

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
