# Peer review of "The Global Burden of Disease of Zoonotic Parasitic Diseases: Top 5 Contenders for Priority Consideration"

_tropicalmed, 2019, doi:10.3390/tropicalmed4010044_

Round 1

Reviewer 1 Report

This was a very interesting, albeit somewhat shortened, discussion on the burden of parasitic zoonoses. The title is descriptive, the references supportive and the text is generally well written. Rightfully so, laboratory-based surveillance methods should be strengthened, but realistic suggestions for improvement (as done for crypto) would also be great, perhaps in a table? Figures of comparative BOD or photos for emphasis may be ideal. Towards the end of the MS, actualization for why these particular pathogens remain neglected would be desirable. In general, when abbreviations are listed, please use consistently. Minor comments follow.

In the keywords, since burden of disease is already listed, the abbreviation of BOD could be excluded and replaced potentially with parasites.

Line 25 AI could also be added.

Line 53 alludes to a critique of the Hotez MS, but info is lacking so suggest some addition or modification as is.

Line 69 cites CDC info, but if it is the US agency, should be the Centers for Disease Control and Prevention.

Line 87 can use INI.

Line 91 the term ‘in’ Asia lacking?

Line 102 seems like a run-on sentence which can be broken up for emphasis to the reader.

Line 125 the Genus is placed in italics here but not in reference to same in relevant lines above.

Line 139 should this reference to the generic be italicized?

Line 150 ‘all sorts of complications’ sounds a tad vernacular and could be tightened.

Line 162 the term ‘of’ missing?

Line 162 ‘Oceania’ capitalized?

Author Response

Thank you for your response and apologies for the late response. I have added an illustrative figure as well as summary tables to aid readers navigate through the paper. I would like to clarify what you mean regarding the actualization of why these particular pathogens remain neglected?

I have taken your comments on board and have addressed them all. Specifically I have specified briefly the critique of Hotez MS with citation however any more than this I feel would be tangential to the scope of the work.

Again, I thank you for your time and your fair critique! Please find attached my amended MS.

- Konrad

Reviewer 2 Report

This manuscript doesn't include the up to date references and was careless written, too many long and repeat sentences,  didn't put references in many places, suggest to  modify again and resubmit.  Some suggestions in the beginning of the manuscript listed as follows: 

1.       Change title “The Global Burden of Disease of Zoonotic Parasites: 2 Top 5 Contenders for Priority Consideration.”  To “ The Global Burden of Zoonotic Parasitic Disease: Top 5 Contenders for Priority Consideration”

2.       Line 21, Change “Burden of Disease; BOD” to “Burden of Disease (BOD)”

3.       Line 37, reference #?

Author Response

My apologies for the late response and for your experience with my initial submission. I have taken your suggestions on board as well as the other reviewers' suggestions and amended the manuscript accordingly. 

Regarding the lack of up to date references and repeat sentences, which ones were these specifically?

Thank you for taking the time to review my submission. Please find attached the amended manuscript.

- Konrad

Round 2

Reviewer 2 Report

Answered all my questions, better than previous version!

Author Response

Regarding the required changes:
1) Figure removed. Agreed.
2) I have addressed this and amended the MS.
3) Checked the spelling of all scientific names, thank you for pointing this out to me!
4) Fixed this in the MS - especially in the schistosomiasis section.
5) Italicized scientific names and tried to make the capitalization consistent in the article titles.
